# Mistreatment from Multiple Sources: Interaction Effects of Abusive Supervision, Coworker Incivility, and Customer Incivility on Work Outcomes

**DOI:** 10.3390/ijerph18105377

**Published:** 2021-05-18

**Authors:** Yuhyung Shin, Won-Moo Hur, Seongho Kang

**Affiliations:** 1School of Business, Hanyang University, 17 Haengdang-dong, Seongdong-gu, Seoul 04763, Korea; yuhyung@hanyang.ac.kr; 2College of Business Administration, Inha University, 100 Inha-ro, Michuhol-gu, Incheon 22212, Korea; wmhur@inha.ac.kr; 3College of Business, Chosun University, 309 Pilmundae-ro, Dong-gu, Gwangju 61452, Korea

**Keywords:** abusive supervision, coworker incivility, customer incivility, emotional exhaustion, job performance

## Abstract

Despite the large body of research on workplace mistreatment, surprisingly few studies have examined the interaction effect of multiple interpersonal stressors on employee outcomes. To fill this gap, our research aimed to test the moderating effects of coworker incivility and customer incivility on the relationship between abusive supervision, emotional exhaustion, and job performance. Analyses conducted on 651 South Korean frontline service employees revealed that abusive supervision exerted a significant indirect effect on job performance through emotional exhaustion. Customer incivility strengthened the positive relationship between abusive supervision and emotional exhaustion, as well as the indirect effect of abusive supervision on job performance through emotional exhaustion. Our post hoc analysis demonstrated a three-way interaction between abusive supervision, coworker incivility, and customer incivility; the relationship between abusive supervision and emotional exhaustion was significantly positive only when coworker incivility was high and customer incivility was low. We discuss the implications of our findings for theory and practice.

## 1. Introduction

Frontline service employees (FSEs) often play a “punching bag” role in organizations. They are the ones who directly receive complaints from customers. At the same time, their work behaviors are observed and evaluated by insiders such as supervisors and coworkers, thereby making them vulnerable to mistreatment by insiders. Despite mounting evidence that FSEs deal with multiple interpersonal stressors [1,2,3,4,5], surprisingly few studies have explored the joint effect of multiple interpersonal stressors on FSEs’ work outcomes. To fill this gap, our research aimed to examine the relationship between FSEs’ multiple interpersonal stressors, emotional exhaustion, and job performance.

Drawing on the findings suggesting that supervisors, coworkers, and customers are the most common sources of interpersonal stress experienced by FSEs [3,6], we hypothesized abusive supervision (supervisors’ hostile verbal and nonverbal behavior), coworker incivility (discourteous behavior by coworkers), and customer incivility (discourteous behavior by customers) to be key interpersonal stressors for FSEs. Prior research has demonstrated the deleterious effects of abusive supervision [1,7,8], coworker incivility [3,6], and customer incivility [1,3,4,9,10,11,12,13] on FSEs’ work outcomes. This stream of research has identified emotional exhaustion (feeling emotionally fatigued and drained) as a mediator that translates the negative effect of abusive supervision into job performance [1,14]. We sought to replicate the mediating effect of emotional exhaustion on the abusive supervision–job performance relationship using a South Korean FSE sample. Thus, the first objective of our research was to test this mediating effect.

Research on multiple interpersonal stressors has further demonstrated that abusive supervision, coworker incivility, and customer incivility independently contribute to increased emotional exhaustion and decreased performance [1,3]. Although this line of research illuminates how different interpersonal stressors affect FSEs’ work outcomes, the interplay between multiple interpersonal stressors has rarely been studied. Given that FSEs are surrounded by multiple interpersonal stressors that operate concurrently [1,3,5,6], it is necessary to elucidate how different interpersonal stressors interact in predicting FSEs’ emotional exhaustion and job performance. Our investigation of the interaction effects of multiple interpersonal stressors could provide a nuanced and comprehensive understanding of the relative roles of different interpersonal stressors. Thus, the second objective of our research was to examine the moderating effects of coworker incivility and customer incivility on the relationship between abusive supervision, emotional exhaustion, and job performance.

## 2. Theoretical Background and Hypothesis Development

Workplace mistreatment research has identified abusive supervision as a common form of mistreatment perpetrated by supervisors. It is defined as “the sustained display of hostile verbal and nonverbal behaviors, excluding physical contact” [15] (p. 178) and exemplified by behaviors such as ridiculing or ignoring the target, making negative comments about the target, not giving credit for jobs that require much effort, and breaking promises. Meanwhile, workplace incivility refers to “low intensity deviant behavior with ambiguous intent to harm the target, in violation of workplace norms for mutual respect” [16] (p. 457). Coworker incivility and customer incivility are rude, discourteous behaviors (e.g., ignoring the target and speaking in a rude manner) instigated by coworkers and customers, respectively. Distinct from aggressive behaviors (e.g., violence, bullying, and deviant behavior), abusive supervision, coworker incivility, and customer incivility are mild forms of workplace mistreatment and have an unclear intent to harm the target [1,17,18]. However, since they are pervasive in the workplace, constant exposure to these interpersonal stressors is detrimental to occupational health and performance [19,20]. Of these three groups of interpersonal stressors, abusive supervision is posited to have an indirect effect on FSEs’ job performance through emotional exhaustion. Job performance refers to the extent to which an employee fulfills their job duties and requirements [21]. Emotional exhaustion, a core dimension of burnout, is defined as a state of feeling emotionally drained [22]. We claim that abusive supervision has adverse effects on employee outcomes because employees are strongly influenced by supervisory behaviors [23]. Because supervisors have the authority, power, and resources to make human resource decisions for employees [1], supervisory mistreatment can have a strong impact on employees’ emotional exhaustion and job performance [23].

### 2.1. Mediating Relationship between Abusive Supervision, Emotional Exhaustion, and Job Performance

Empirical findings have denoted the mediating effect of emotional exhaustion on the abusive supervision–employee performance relationship [1,3,14]. In line with these findings and the conservation of resources (COR) theory [24], we propose that abusive supervision negatively affects FSEs’ job performance through emotional exhaustion. In the COR framework, resources are defined as the objects, personal characteristics, or conditions that are valued by the individual [24]. According to this framework, as individuals are inclined to acquire and preserve their valued resources, deficiency in or loss of resources becomes a key stressor to them. Applying this theory to employee–supervisor dyads, employees who interact with abusive supervisors have difficulty gaining access to valued resources such as good relationships with supervisors, recognition from supervisors, promotions, and pay raises [1]. Deprived of important social and job resources, FSEs under abusive supervision become incapable of coping with interpersonal conflict with the supervisor, and thereby feel emotionally exhausted. Exposure to one interpersonal conflict causes employees to experience another interpersonal conflict through the COR processes [25]. Abusive supervision is a resource-depleting condition for FSEs since it may threaten their current working conditions and decrease psychological resources (e.g., self-esteem and self-efficacy) [26]. Moreover, dealing with abusive supervisors is often accompanied by negative emotions (e.g., anger, worry, and fear) [1,27]. Because FSEs interacting with an abusive supervisor may not retaliate owing to the inequality of their relationship and fear of a potential job loss, they often suppress the expression of negative emotions toward the supervisor. Such emotion regulation expends considerable emotional resources, which results in emotional exhaustion.

Emotional exhaustion, in turn, is anticipated to impair FSEs’ job performance. The negative link between emotional exhaustion and job performance has been well documented [28,29,30]. As predicted by COR theory, emotionally exhausted employees lack the mental and physical resources required to perform their jobs. Emotional exhaustion hinders FSEs from concentrating on core work activities and attenuates their motivation to perform well, which leads to diminished job performance [30]. Integrating the proposed relationships, abusive supervision was proposed to contribute to FSEs’ emotion exhaustion by depleting their mental and emotional resources, which in turn undermines their job performance. This line of reasoning led to the following mediation hypothesis:

**Hypothesis** **1** **(H1).**
*Abusive supervision has a negative indirect effect on FSEs’ job performance through emotional exhaustion.*


### 2.2. Moderating Effects of Coworker and Customer Incivility on the Abusive Supervision–Emotional Exhaustion Relationship

Grounded in the contention that FSEs are influenced by multiple interpersonal stressors in the workplace [1,3], coworker and customer incivility interact with abusive supervision to predict emotional exhaustion. Hobfoll [24], in his COR model, postulated a loss spiral in which a resource loss increases the vulnerability to a further resource loss. According to this theory, abusive supervision creates stressful situations for FSEs. As dealing with the abusive supervisor exhausts their mental and emotional resources, they become unable to cope with any other interpersonal stressor, such as coworker incivility. COR theory further asserts that a resource loss stemming from an interpersonal stressor can be offset by social support [24]. That is, coworker emotional support provides employees suffering from abusive supervision with the resources necessary to cope with such a stressor. In particular, emotional support from coworkers alleviates feelings of rejection and incompetence arising from abusive supervision. Indeed, coworker support has been found to attenuate the adverse effects of interpersonal stressors on work outcomes [26,31,32,33]. Meanwhile, employees experiencing coworker incivility lack the social resources that protect them against abusive supervision, which worsens the deleterious effect of abusive supervision. Further, interactions with both abusive supervisors and uncivil coworkers exacerbate FSEs’ emotional exhaustion, since they feel isolated at work and their relatedness needs remain unmet. As such, the positive relationship between abusive supervision and emotional exhaustion should be more pronounced when coworker incivility is high.

**Hypothesis** **2** **(H2).**
*Coworker incivility moderates the positive relationship between abusive supervision and emotional exhaustion such that this relationship is more pronounced when coworker incivility is high than when it is low.*


Customer incivility can also strengthen the abusive supervision–emotional exhaustion association. As suggested by COR theory, abusive supervision drains FSEs’ mental and emotional resources, which leaves them susceptible to customer incivility. Supervisors guide FSEs’ work behaviors by clarifying and training desired service behaviors and emotional display rules [34,35]. Supervisors are also responsible for teaching FSEs to handle uncivil customers. When FSEs interact with an abusive supervisor, they have difficulty receiving important instructions and guidance from the supervisor, which precludes them from coping with customer incivility. Dealing with abusive supervisors and uncivil customers at the same time is a devastating situation for FSEs owing to the unequal status between the FSEs and the other parties. As noted earlier, supervisors have higher organizational status and greater power than FSEs [1], which dissuades FSEs from venting their negative emotions toward the supervisor. Similarly, expressing aggression toward customers is unacceptable for FSEs because of the “customer is the king” policy [36,37,38]. Therefore, FSEs tend to suppress their negative emotions and fake positive emotions when interacting with abusive supervisors and uncivil customers [1]. This emotional regulation depletes FSEs’ emotional resources and thus induces a state of emotional exhaustion. Based on this logic, customer incivility is posited to function as a moderator that escalates the resource-depleting effect of abusive supervision. We therefore expected the relationship between abusive supervision and emotional exhaustion to be more profound when customer incivility is high than when it is low:

**Hypothesis** **3** **(H3).**
*Customer incivility moderates the positive relationship between abusive supervision and emotional exhaustion such that this relationship is more pronounced when customer incivility is high than when it is low.*


### 2.3. Moderating Effects of Coworker and Customer Incivility on the Abusive Supervision–Emotional Exhaustion–Job Performance Relationahip

Synthesizing the aforementioned hypotheses, we propose moderated mediation relationships in which coworker incivility and customer incivility moderate the indirect effect of abusive supervision on job performance through emotional exhaustion. These relationships are bolstered by COR theory, which asserts that the resource loss triggered by a work stressor renders employees vulnerable to another stressor, thus aggravating its resource-depleting effect [24]. Experiencing more than one interpersonal stressor accelerates the resource-depletion process of the stressor, resulting in a high level of emotional exhaustion. More specifically, employees who are faced with abusive supervision and coworker incivility expend a substantial amount of resources in coping with the two stressors. In such a state of emotional exhaustion, FSEs lack the psychological resources for task accomplishment, thereby leading them to perform their jobs poorly. Likewise, the resource depletion triggered by abusive supervision is accelerated by customer incivility. Because FSEs need to regulate their emotions toward supervisors and customers, they tend to consume their psychological resources quickly and thus feel emotionally fatigued. As a consequence, FSEs are deprived of the psychological resources required for task activities, which in turn undermines their job performance. Taken together, we formulated the following moderated mediation hypotheses:

**Hypothesis** **4** **(H4).**
*Coworker incivility moderates the indirect effect of abusive supervision on job performance through emotional exhaustion such that this indirect effect is more pronounced when coworker incivility is high than when it is low.*


**Hypothesis** **5** **(H5).**
*Customer incivility moderates the indirect effect of abusive supervision on job performance through emotional exhaustion such that this indirect effect is more pronounced when customer incivility is high than when it is low.*


Figure 1 illustrates the research model used in our work.

## 3. Method

### 3.1. Sample and Procedure

Our sample consisted of South Korean FSEs in various service organizations (e.g., airlines, banks, hospitality, retail, etc.) who were recruited by an online survey company. Online survey panels are known to be a reliable source of access to diverse samples [39,40]. We invited 721 FSEs to participate in our research by emailing them an informed consent form along with an online survey link. Among the FSEs who completed the prescreening questionnaire, FSEs in nonmanagerial positions whose job tenure was longer than one year were invited to our research. This was because it takes time for entry-level FSEs to experience all three interpersonal stressors. Employees who completed the online survey received USD 3 as a reward for participation. A total of 651 employees participated in our research (response rate = 77.8%). Sixty-four percent of the participants were women. The average age of the participants was 35.69 (SD = 8.51) years, ranging from 21 to 54 years. The education level of the participants varied: high school diploma (27.8%), bachelor’s degree (70.2%), and master’s degree/Ph.D. (2.0%). The participants, on average, reported 4.68 (SD = 4.21) years of experience in their current job (job tenure).

### 3.2. Measures

As the original survey items were written in English, they were translated to Korean and then back-translated and validated by bilingual scholars [41]. With the exception of coworker and customer incivility, all the other variables were measured on a five-point Likert-type scale (1 = strongly disagree, 5 = strongly agree). Responses for coworker and customer incivility were made on a scale ranging from 1 (*never*) to 5 (*very often*) (see Table 1).

Abusive supervision was assessed using Tepper’s [15] six-item abusive supervision short version scale (*α* = 0.93). Coworker incivility was measured with four items adapted from the Sliter et al. [5] scale (*α* = 0.91). Customer incivility was gauged with 10 items from Wilson and Holmvall’s [42] scale (*α* = 0.94). Emotional exhaustion was assessed using four items from Maslach and Jackson’s [22] scale (*α* = 0.84). Job performance was evaluated with four items from Williams and Anderson’s [21] in-role performance scale (*α* = 0.90).

We controlled for the participants’ gender, age, job tenure, positive affectivity, and negative affectivity in consideration of their potential confounding effects on emotional exhaustion [43,44,45] and job performance [44,46,47]. Positive affectivity (*α* = 0.90) and negative affectivity (*α* = 0.89) were measured with the 10-item International Positive Affect and Negative Affect Schedule Short Form [48].

## 4. Results

### 4.1. Test of Reliability, Validity, and Common Method Variance

Table 2 presents the means, standard deviations, Cronbach’s alpha coefficients, and correlations of the study variables. The Cronbach’s alphas for the scales ranged from 0.84 to 0.94, demonstrating a high level of reliability [49]. To evaluate the convergent and discriminant validity, we conducted a confirmatory factor analysis using M-plus 8.4. As reported in Table 1, the suggested eight-factor model (i.e., abusive supervision, coworker incivility, customer incivility, emotional exhaustion, job performance, positive affectivity, and negative affectivity) exhibited an acceptable fit in an absolute sense (χ ^2^_(644)_ = 1845.61; *p* < 0.05; comparative fit index (CFI) = 0.93; Tucker–Lewis index (TLI) = 0.93; root mean square error of approximation (RMSEA) = 0.05; standardized root mean square residual (SRMR) = 0.04). Furthermore, the eight constructs displayed a sufficient level of composite reliability ranging from 0.85 to 0.94 (see Table 2). Additionally, we assessed the discriminant validity among the constructs based on Fornell and Larcker’s [50] procedure. Table 2 shows that all average variances extracted (AVEs) exceeded the squared correlations between the target construct and each of the other constructs.

Because we relied on self-reported measures, we explored the possibility that the participants’ responses were affected by common method variance (CMV). Podsakoff, MacKenzie, and Podsakoff [51] claimed that researchers can reduce CMV using statistical and procedural remedies. Based on their recommendation, we employed procedural remedies by ensuring the anonymity and confidentiality of survey responses and improving the wording of survey items. Additionally, we conducted Harman’s one-factor analysis as a statistical remedy [51]. All measures of the goodness of fit indicated a worse fit for the one-factor model than for the original measurement model (χ^2^_(665)_ = 12009.57; *p <* 0.05, CFI = 0.36, TLI = 0.23, RMSEA = 0.16, SRMR = 0.17). The standardized factor loadings of all items were below 0.50 for the latent common method factor, and only 5.02% of the factor loadings of the manifest variables on the latent common method factor were significant at the 5% level. Based on these findings, we concluded that our data were not affected by CMV.

### 4.2. Hypothesis Testing

We tested our hypotheses in two steps. First, we assessed a simple mediation model to test Hypothesis 1. Second, to test the moderation (Hypotheses 2 and 3) and moderated mediation (Hypotheses 4 and 5) effects, we conducted three-way moderated mediation analysis. Prior to the main analyses, all continuous variables were mean-centered [52]. To analyze the mediation, three-way moderation, and three-way moderated mediation effects, we used a PROCESS macro for SPSS [53].

Hypothesis 1 proposed that emotional exhaustion would mediate the negative relationship between abusive supervision and job performance. We tested this hypothesis using a bootstrapping (N = 5000) procedure, a statistical resampling method that estimates the standard deviation of a model from a sample [53]. The results showed that, controlling for gender, age, job tenure, positive affectivity, and negative affectivity, the negative indirect effect of abusive supervision on and job performance through emotional exhaustion was significant (*b* = −0.052, 95% confidence interval [CI] = [−0.084, −0.022]). Moreover, when emotional exhaustion was included in the model, the direct effect of abusive supervision on job performance was no longer statistically significant (*b* = −0.037, 95% CI = [−0.089, 0.014]), suggesting full mediation (see Table 3). These findings lend support to Hypothesis 1.

In Hypotheses 2 and 3, we predicted moderation by coworker and customer incivility of the relationship between abusive supervision and emotional exhaustion, respectively. Contrary to our prediction, coworker incivility did not moderate the relationship between abusive supervision and emotional exhaustion (*b* = −0.04, *p >* 0.05). Thus, Hypothesis 2 was not supported (see Table 4). As depicted in Table 4, customer incivility strengthened the positive relationship between abusive supervision and emotional exhaustion (*b* = 0.13, *p <* 0.01). In addition, a follow-up simple slope analysis (plotting simple slopes at ±1 SD of the moderator) demonstrated that the positive relationship between abusive supervision and emotional exhaustion was more pronounced among employees who reported average and high levels of customer incivility (average: *b* = 0.11, 95% CI = [0.03, 0.18]; high: *b* = 0.22, 95% CI = [0.13, 0.30]) (see Figure 2). In contrast, abusive supervision was not associated with emotional exhaustion for low-level customer incivility (low: *b* = −0.00, 95% CI = [−0.10, 0.10]). These findings provide support for Hypothesis 3.

Hypotheses 4 and 5 postulated moderated mediation relationships in which coworker and customer incivility, respectively, would strengthen the indirect effect of abusive supervision on job performance through emotional exhaustion. To test these hypotheses, we estimated the conditional indirect effect of abusive supervision and job performance through emotional exhaustion depending on the levels of coworker and customer incivility. As shown in Table 4, the moderated mediation of coworker incivility was not significant (*b* = 0.014, 95% CI = [−0.021, 0.045]). Thus, Hypothesis 4 was not supported. However, the conditional indirect effect of abusive supervision on job performance via emotional exhaustion was strengthened by customer incivility (*b* = −0.044, 95% CI = [−0.076, −0.010]). More precisely, the negative indirect effect of abusive supervision on job performance was significant for average and high levels of customer incivility (average: *b* = −0.036, 95% CI = [−0.068, −0.007]; high: *b* = −0.074, 95% CI = [−0.115, −0.073]). Conversely, when customer incivility was low, the negative indirect effect of supervisor incivility on service performance was not significant (low: *b* = 0.001, 95% CI = [−0.041, 0.039]), thereby supporting Hypothesis 5.

### 4.3. Post Hoc Analysis

Based on a study pointing to the joint effect of coworker and customer incivility [5], we tested a three-way interaction between abusive supervision, coworker incivility, and customer incivility on emotional exhaustion. As illustrated in Figure 3, the three-way interaction was significant (*b* = 0.08, *p <* 0.05). We further plotted this three-way interaction in Figure 4. Abusive supervision had a significant negative relationship with emotional exhaustion only when coworker incivility was high and when customer incivility was low, suggesting a buffering effect of low customer incivility. In the other three conditions, the association between abusive supervision and emotional exhaustion was not significant.

## 5. Discussion

The present study aimed to assess the mediating effect of emotional exhaustion on the abusive supervision–job performance relationship and the moderating effects of coworker incivility and customer incivility on this relationship. As predicted, abusive supervision had a significant indirect effect on FSEs’ job performance through emotional exhaustion. Of the two types of incivility, only customer incivility exerted a moderating effect on emotional exhaustion. The negative relationship between abusive supervision and job performance was more pronounced when customer incivility was high than when it was low. Customer incivility further moderated the indirect effect of abusive supervision on job performance through emotional exhaustion. Our post hoc analysis revealed a significant interaction effect between abusive supervision, coworker incivility, and customer incivility on emotional exhaustion such that the relationship between abusive supervision and emotional exhaustion became negative when coworker incivility was high and when customer incivility was low.

### 5.1. Theoretical Implications

We found that emotional exhaustion significantly mediated the negative relationship between abusive supervision and job performance, even after controlling for psychological resources such as positive affectivity. This result is consistent with prior findings highlighting the mediating role of emotional exhaustion in the abusive supervision–employee performance relationship [1,3,14]. The mediating effect of emotional exhaustion also corroborates the COR proposition that interpersonal stressors are a primary condition that drains employees’ psychological resources. Abusive supervision signifies rejection by and isolation from the supervisor [54], leading employees to believe that they cannot obtain important resources in the workplace. Such a resource loss leads to a loss of psychological resources (e.g., positive emotions, self-esteem, self-efficacy), causing employees to feel emotionally exhausted. Because these employees lack the mental and emotional resources for task accomplishment, they tend to perform poorly. As such, by identifying emotional exhaustion as a key mediator translating the effect of abusive supervision into job performance, our findings offer theoretical accounts for how abusive supervision negatively impacts FSEs’ job performance.

The significant moderating effect of customer incivility on the abusive supervision–job performance relationship and the abusive supervision–emotional exhaustion–job performance relationship also validates the notion of a loss spiral proposed by COR theory. According to this theory, employees experiencing a resource loss are vulnerable to further losses of resources [24]. In support of this premise, we found that the detrimental effect of abusive supervision on work outcomes was aggravated by exposure to a high level of customer incivility. This is because the resource depletion caused by abusive supervision hinders employees’ coping with a different interpersonal stressor. Mistreatment occurring in unequal relationships is more exhausting owing to the increased emotional regulation [1]. Our investigation of the interaction effect of multiple interpersonal stressors provided novel insight into the interplay between abusive supervision and customer incivility by revealing that exposure to these two interpersonal stressors is most harmful to FSEs’ work outcomes. Thus, complementing prior research on the independent effects of multiple interpersonal stressors on work outcomes, our study is one of the first to disentangle the dynamics between different interpersonal stressors.

Our research adds to the extant literature on cross-cultural abusive supervision by examining the effect of abusive supervision in the Korean context. Our findings are consistent with the meta-analytic finding that abusive supervision is more prevalent in Asian countries than in the United States [55]. The average level of abusive supervision perceived by Korean employees in our research (M = 2.09) was similar to those in China (M = 2.06), the Philippines (M = 2.17), and Taiwan (M = 2.13), but was higher than the abusive supervision level in the United States (M = 1.68). This is attributable to the high power distance of Asian countries [56]. In high-power-distance cultures, abusive supervision is quite normative because employees respect and obey to supervisors [57]. It should be noted that although abusive supervision is more common in Asian countries than in Western countries, the deleterious effect of abusive supervision is more severe in Western countries. Zhang and Liao’s [57] meta-analysis revealed that the negative effect of abusive supervision on employee work outcomes is stronger in North America than in Asia. This may be because individuals in low-power-distance cultures are less tolerant and more resentful of abusive supervision [57]. Thus, the negative indirect effect of abusive supervision on job performance through emotional exhaustion might have been more pronounced if this effect were studied in low-power-distance cultures.

Our post hoc analysis results contribute to a nuanced understanding of different interpersonal stressors. The lowest levels of emotional exhaustion were observed when both coworker and customer incivility were low. However, the presence of high coworker incivility and high customer incivility did not strengthen the relationship between abusive supervision and emotional exhaustion. One interesting finding is that abusive supervision reduced emotional exhaustion when coworker incivility was high and when customer incivility was low. This finding indicates that despite being surrounded by uncivil coworkers, if FSEs experience low customer incivility, they do not need to consume emotional resources to handle difficult customers. Such a surplus of emotional resources would enable FSEs to cope with abusive supervision. Taken together, such a three-way interaction, coupled with the significant two-way interaction between abusive supervision and customer incivility, suggests that customer incivility is an important boundary condition that affects the abusive supervision–emotional exhaustion relationship. While high customer incivility aggravates the deleterious effect of abusive supervision, low customer incivility can buffer this effect.

Contrary to our prediction, coworker incivility failed to moderate the abusive supervision–job performance relationship and abusive supervision–emotional exhaustion–job performance relationship, which is consistent with prior findings that coworker incivility exerts a weaker effect on employee outcomes than abusive supervision and customer incivility [3,6,58]. The nonsignificant effect of coworker incivility is attributable to the equal status between employees and coworkers. As retaliation for uncivil behaviors and expression of negative emotions are perceived to be less risky in employee–coworker relationships, coworker incivility is less threatening and stressful to FSEs than abusive supervision or customer incivility, thereby resulting in less emotional exhaustion compared with the other two interpersonal stressors. This explanation is also endorsed by our post hoc finding that coworker incivility did not play an influential role in the relationship between abusive supervision and emotional exhaustion.

### 5.2. Practical Implications

Based on the findings that abusive supervision is detrimental to FSEs’ work outcomes and that customer incivility exacerbates such negative effects, organizations that strive to promote FSEs’ performance should be cognizant of the potential effects of interpersonal stressors on employee outcomes. Scholars recommend organizational policies that control abusive supervision [1]. An example of such policies would be selecting and promoting individuals with desirable personality traits and leadership skills [23]. In addition, penalizing supervisors who abuse their subordinates can prevent or reduce incidence of abusive supervision. Organizations are also advised to provide supervisors with feedback regarding subordinates’ evaluation of their supervisory behaviors and to implement training programs that warn supervisors against the hazards of abusive supervision and teach them proper supervisory behaviors [1,23,59].

Unlike incivility instigated by organizational members, customer incivility is difficult to control. It is unrealistic to teach customers etiquette. Instead, fortifying FSEs’ coping capabilities would be a more feasible solution. Emotion regulation or mindfulness training can help FSEs to relieve stress arising from difficult customer encounters and therefore become better able to cope with such stressors [38,45]. Customer incivility researchers also recommend short breaks after an episode of customer incivility [1,60] to enable FSEs to recharge the emotional resources depleted during customer interactions.

### 5.3. Study Limitations and Directions for Future Research

Despite its theoretical and practical implications, our study is not without limitations. First, because we employed cross-sectional data, we could not ascertain the causal relationships between abusive supervision, emotional exhaustion, and job performance. There is empirical evidence that subordinates’ poor performance results in abusive supervision by evoking supervisors’ anger [61]. Likewise, emotionally exhausted employees can be easy targets of abusive supervision. Considering such reverse causality, causal inferences regarding the present findings should be made with caution. To resolve this issue, future research needs to use more rigorous research designs, such as longitudinal panel designs.

Second, the use of self-reported data is another limitation of our research. Although we used several procedural and statistical remedies to reduce CMV, the relationship between abusive supervision, emotional exhaustion, and job performance might have been overestimated because of CMV. In addition, self-reported job performance is susceptible to rater biases (e.g., social desirability and evaluation apprehension). In our data, job performance exhibited a negatively skewed distribution (M = 3.92), indicating respondents’ leniency in evaluating their own performance. This problem could be remedied by using more objective measures (e.g., supervisors’ or coworkers’ ratings of focal employees).

Third, while we focused on the interaction effects of multiple interpersonal stressors, employees’ reactions to interpersonal stressors can vary depending on their personal resources. Employees’ resilience can buffer the negative effects of abusive supervision and customer incivility [1]. Given that individuals’ personal resources help them to cope with stressors [24], the buffering role of such personal resources in multiple interpersonal stressors comprises a future research agenda.

Finally, although our findings provide insights into the roles of multiple interpersonal stressors among Korean FSEs, we did not conduct a cross-cultural comparison. As noted earlier, the relationship between abusive supervision and employee work outcomes can be affected by cultural characteristics. Thus, the effect of multiple interpersonal stressors can be better captured in cross-cultural research. We encourage future researchers to explore the interplay between abusive supervision, coworker incivility, and customer incivility using multinational samples.

## 6. Conclusions

Our research aimed to investigate the roles of coworker incivility and customer incivility in the relationship between abusive supervision, emotional exhaustion, and job performance. By demonstrating that customer incivility (not coworker incivility) accentuated the deleterious effect of abusive supervision on employee outcomes, our research broadens our understanding of the interplay between multiple interpersonal stressors. Furthermore, by unveiling the relative importance of different types of incivility in coping with abusive supervision, our findings provide a nuanced understanding of mistreatment from different sources. Future research on personal resources as a buffer against interpersonal stressors will enrich the insights gained from the present study.

## Figures and Tables

**Figure 1 ijerph-18-05377-f001:**
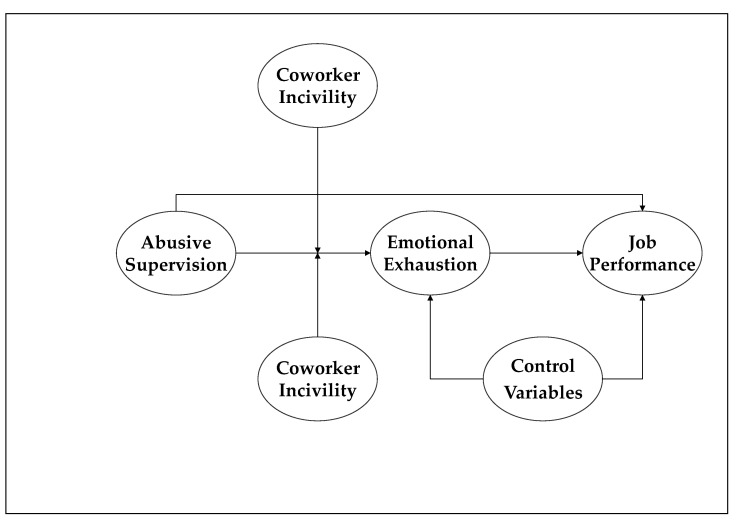
Proposed research model. For parsimony, the control variables are not included in this figure.

**Figure 2 ijerph-18-05377-f002:**
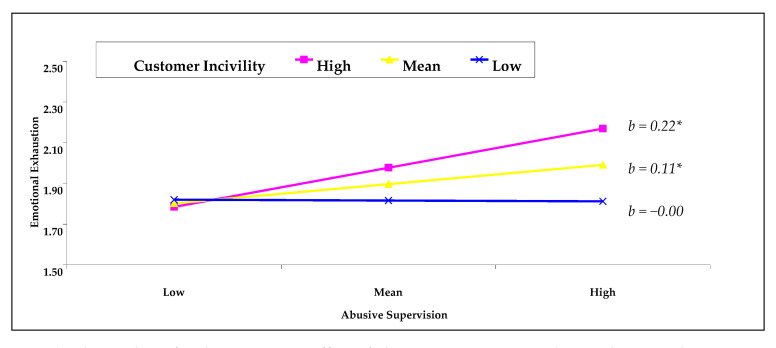
Simple plot analysis for the interaction effect of abusive supervision and coworker incivility on emotional exhaustion. ** p <* 0.05. *b* = unstandardized coefficient.

**Figure 3 ijerph-18-05377-f003:**
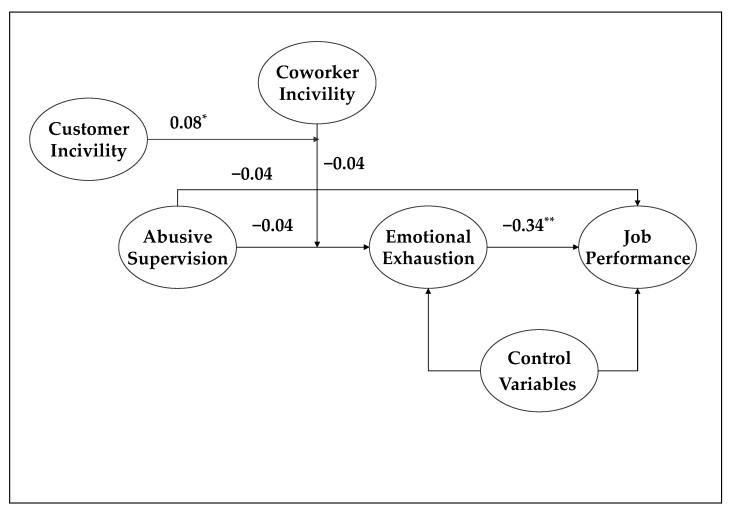
Three-way interaction effect of abusive supervision, coworker incivility, and customer incivility on emotional exhaustion.* *p* < 0.05, ** *p* < 0.01. *b* = unstandardized coefficient.

**Figure 4 ijerph-18-05377-f004:**
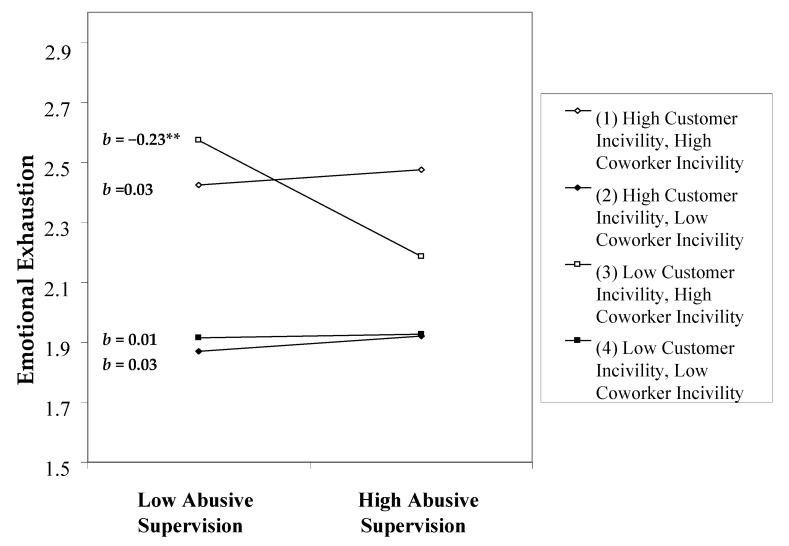
Simple plot analysis for the three-way interaction effect of abusive supervision, coworker incivility, and customer incivility on emotional exhaustion. ** *p* < 0.01*. b* = unstandardized coefficient.

**Table 1 ijerph-18-05377-t001:** Factor analysis results of measurement items.

Construct	Measurement Items	λ ^(c)^
Abusive supervision ^(a)^	My supervisor makes negative comments about me to others.	0.78
	My supervisor gives me the silent treatment.	0.84
	My supervisor expresses anger at me when he/she is mad for another reason.	0.88
	My supervisor is rude to me.	0.85
	My supervisor breaks promises he/she makes.	0.85
	My supervisor puts me down in front of others.	0.83
Coworker incivility ^(b)^	How often do coworkers ignore or exclude you while at work?	0.84
	How often do coworkers raise their voices at you while at work?	0.79
	How often are coworkers rude to you at work?	0.92
	How often do coworkers do demeaning things to you at work?	0.83
Customer incivility ^(b)^	*How often have customers…*	
	…continued to complain despite your efforts to assist them?	0.75
	…made gestures (e.g., eye rolling, sighing) to express their impatience?	0.74
	…grumbled to you about slow service during busy times?	0.87
	…made negative remarks to you about your organization?	0.82
	…blamed you for a problem you did not cause?	0.82
	…used an inappropriate manner of addressing you (e.g., “Hey, you”)?	0.73
	…failed to acknowledge your efforts when you have gone out of your way to help them?	0.83
	…grumbled to you that there were too few employees working?	0.79
	…complained to you about the value of goods and services?	0.80
	…made inappropriate gestures to get your attention (e.g., snapping fingers)?	0.73
Emotional exhaustion ^(a)^	I feel frustrated with my job.	0.56
	I feel used up at the end of the workday.	0.79
	I feel like I am working too hard in my job.	0.87
	I feel like I am at the end of my rope.	0.81
Job performance ^(a)^	I adequately complete assigned duties.	0.84
	I fulfill the responsibilities specified in my job description.	0.89
	I perform the tasks that are expected of me.	0.80
	I meet the formal performance requirements of my job.	0.81
Positive affectivity ^(a)^	Determined	0.79
	Attentive	0.83
	Alert	0.86
	Inspired	0.82
	Active	0.67
Negative affectivity ^(a)^	Afraid	0.78
	Nervous	0.82
	Upset	0.86
	Ashamed	0.82
	Hostile	0.64
χ^2^_(644)_ = 1845.61; *p* < 0.05, CFI = 0.93, TLI = 0.93, RMSEA = 0.05, SRMR = 0.04		

^(a)^ Items measured on a scale ranging from 1 “strongly disagree” to 5 “strongly agree.” ^(b)^ Items measured on a scale ranging from 1 “never” to 5 “very often.” ^(c)^ All factor loadings are significant (*p <* 0.01).

**Table 2 ijerph-18-05377-t002:** Means, standard deviations, and correlations of all variables.

Variables	M	SD	α	CR	AVE	1	2	3	4	5	6	7	8	9	10
1. Gender	0.36	0.48	-	-	-	1									
2. Age	35.69	8.51	-	-	-	−0.12 **	1								
3. Job tenure	4.68	4.21	-	-	-	0.04	0.39 **	1							
4. Positive affectivity	2.51	0.84	0.90	0.90	0.64	0.13 **	0.06	−0.10 **	1						
5. Negative affectivity	2.96	0.92	0.89	0.91	0.68	−0.11 **	−0.22 **	−0.02	0.30 **	1					
6. Abusive supervision	2.09	0.89	0.93	0.93	0.70	0.10 *	−0.06	−0.00	−0.12 **	0.28 **	1				
7. Coworker incivility	2.01	0.87	0.91	0.93	0.76	0.10 **	0.06	0.05	−0.08 *	0.24 **	0.56 **	1			
8. Customer incivility	2.62	0.84	0.94	0.94	0.62	−0.12 **	−0.21 **	0.04	−0.23 **	0.50 **	0.32 **	0.28 **	1		
9. Emotional exhaustion	2.21	0.82	0.84	0.85	0.59	0.09 *	−0.01	0.04	0.21 **	0.26 **	0.24 **	0.39 **	0.23 **	1	
10. Job performance	3.92	0.65	0.90	0.90	0.70	−0.12 **	−0.06	−0.09 *	0.22 **	−0.08 ^†^	−0.16 **	−0.24 **	−0.08 *	−0.47 **	1

^†^*p <* 0.10, * *p* < 0.05, ** *p* < 0.01. CR = composite reliability. AVE = average variance extracted. Gender: 0 = female, 1 = male.

**Table 3 ijerph-18-05377-t003:** Test of the mediating effect of emotional exhaustion on the abusive supervision–job performance relationship.

Path	Effect *(b)*	95% CI_low_	95% CI_high_
**Total Effect**			
Abusive supervision→Job performance	−0.089	−0.145	−0.032
**Direct Effect**			
Abusive supervision→Job performance	−0.037	−0.089	0.014
**Indirect Effect**			
Abusive supervision→Emotional exhaustion→Job performance	−0.052	−0.084	−0.022

Unstandardized coefficients are reported.

**Table 4 ijerph-18-05377-t004:** Test of the interaction effects of abusive supervision, coworker incivility, and customer incivility.

Variables	Emotional Exhaustion		Job Performance	
	*b*	(se)	*b*	(se)
Gender	0.16	(0.06) **	−0.10	(0.05) *
Age	−0.00	(0.04)	−0.00	(0.00)
Job tenure	−0.01	(0.08)	−0.01	(0.01)
Positive affectivity	−0.15	(0.04) **	0.12	(0.03) **
Negative affectivity	0.13	(0.04) **	0.06	(0.03) *
Abusive supervision	−0.04	(0.04)	−0.04	(0.03)
Coworker incivility	0.32	(0.04) **		
Customer incivility	0.06	(0.04)		
Abusive supervision × Coworker incivility	−0.04	(0.03)		
Abusive supervision × Customer incivility	0.13	(0.04) **		
Emotional exhaustion			−0.34	(0.03) **
*R* ^2^	22.3%		25.4%	
*Moderated mediation index*				
Abusive supervision × Coworker incivility→Emotional exhaustion→Job performance: *b* = 0.014, 95% CI = [−0.021, 0.045]				
Abusive supervision × Customer incivility→Emotional exhaustion→Job performance: *b* = −0.044, 95% CI = [−0.076, −0.010]				

** p* < 0.05*, ** p* < 0.01. *b* = unstandardized coefficient.

## Data Availability

The data presented in this study are available on request from the corresponding author.

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
