# Peer review of "Mistreatment from Multiple Sources: Interaction Effects of Abusive Supervision, Coworker Incivility, and Customer Incivility on Work Outcomes"

_ijerph, 2021, doi:10.3390/ijerph18105377_

Round 1

Reviewer 1 Report

This study aimed to test the moderating effects of coworker incivility and customer incivility on the relation between abusive supervision, emotional exhaustion, and job performance. The topic is interesting. The manuscript was well written. However, some comments below need to be addressed.

  1. Why are there some unstandardized coefficients in Figure 1? It should show only the research model of this study.
  2. What are the selection criteria of the sample of this study?
  3. There should be a conclusion of this study. In the conclusion, the main findings and practical implications should be included.

Reviewer 2 Report

The research aimed to test the moderating effects of coworker incivility and customer incivility on the relation between abusive supervision, emotional exhaustion and job performance. Analyses were conducted on 651 South Korean frontline service employees. It revealed that abusive supervision exerted a significant indirect effect on job performance through emotional exhaustion. Customer incivility strengthened the positive relation between abusive supervision and emotional exhaustion, as well as the indirect effect of abusive supervision on job performance through emotional exhaustion. The post-hoc analysis demonstrated a three-way interaction between abusive supervision, coworker incivility and customer incivility. The Authors discussed the implications of their findings for theory and practice.

The Authors created for us the theoretical background, that can be used for future research, too, due to wide range of it.

The methodology used in submision help to fill the scientist gap in researched  area. The sample consisted of South Korean FSEs in various service organizations. A total amount of employees that participated in research was 651. The research is complete, comprehensive and interesting. The results are clearly presented and answered the formulated hypotheses. 

I'm impressed by oryginality of research and mature conclusions.

The one point that can improve the value of article is the comparision of gained effects with desk research, carried in other countries (multinational comparisions).

Reviewer 3 Report

The data presented in Table 2 is not presented in a way that is "normal" for the way relationships are traditionally presented. The diagonal should be how the variable correlates with itself and is generally represented with a -. However, it appears from the note that the authors have done something different here. I would like more clarification on the presentation of this data.
